# Long-term changes in body image after bariatric surgery: An observational cohort study

**Laurène Bosc**[1], **Flore Mathias**[1☯], **Maud Monsaingeon**[1☯], **Caroline Gronnier**[2,3☯], **Emilie Pupier**[1☯], **Blandine Gatta-Cherifi**[1,3,4]*

**1** Endocrinology Department, Bordeaux University Hospital, Pessac, France, **2** Digestive Surgery Department, Bordeaux University Hospital, Pessac, France, **3** University of Bordeaux, Bordeaux, France, **4** Neurocentre Magendie, Physiopathologie de la Plasticité Neuronale, University of Bordeaux, U1215, Bordeaux, France

☯ These authors contributed equally to this work.
* blandine.gatta-cherifi@chu-bordeaux.fr

**Data Availability Statement:** All relevant data are within the manuscript and its Supporting Information files.

## Abstract

### Background

While body image improves in the first few months after surgery, data on long-term changes in body image after bariatric surgery are scarce and contradictory.

### Methods

We assessed body image through the Stunkard Figure Rating Scale and the Multidimensional Body-Self Relations Questionnaire-Appearance Scale, which measures appearance evaluation and orientation, overweight preoccupation, and self-classified weight. Surveys were conducted before surgery and at regular intervals until 5 years after bariatric surgery.

### Results

61 patients were included in the study. No patients were lost to follow-up until 18 months after bariatric surgery. At 5 years, there were 21 patients (34%) lost to follow-up. We detected an overall improvement in body image until 12–18 months post-surgery. Scores declined after 5 years post-surgery but were still higher than preoperative evaluations. Overweight preoccupation did not change throughout the follow-up period. There was a positive correlation between body weight lost and appearance evaluation. There was also a positive correlation between weight loss and the Body Areas Satisfaction Scale. There was a negative correlation between weight loss and overweight preoccupation. Appearance orientation and self-classified weight were not correlated with weight loss.

### Conclusions

Body image improved after bariatric surgery but was not maintained for all 5 years after surgery.

**Funding:** The authors received no specific funding for this work.

**Competing interests:** The authors have declared that no competing interests exist.

## Introduction

Body image dissatisfaction is prevalent among the general population [1] and common in patients suffering from obesity [2], women [3], White, and those with a 'Western lifestyle' [1]. In patients with obesity, early-onset obesity, obesity-related stigmas, eating disorders, underlying depression, yo-yo dieting effects, and body image distortion are correlated with poorer body image satisfaction [4, 5].

Bariatric surgery (BS) is the most efficient treatment for morbid obesity. Depending on the procedure, it allows for 13–27% body weight loss for up to 15 years post-surgery and reduces obesity mortality by 5% [6]. It also has a dramatic positive effect on obesity-related comorbidities: especially type 2 diabetes but also cardiovascular diseases including hypertension, obstructive sleep apnea-hypopnea syndrome and also non-alcoholic steatohepatitis, osteoarticular disorders [7].

Health is the primary reason reported by patients for seeking bariatric surgery but patients also report a strong desire for surgery to change the appearance of their body [8, 9]. Indeed, approximately one in five bariatric surgery patients identifies appearance concerns as the primary motivator for bariatric surgery [10]. While the weight loss and metabolic outcomes of bariatric surgery have been well-documented, the evolution of body image following bariatric surgery is less investigated [11]. Most studies report only short-term results and the literature on post-operative body image changes and weight outcome is mixed [12–15]. Therefore, if general improvement is typically observed among adults, some also reported improvements in specific body image domains only. The mixed literature may perhaps be related to methodological factors including the variety of body image measures used (more than 20 different body measures which have not been validated most of the time for body image assessment after bariatric surgery) which tap different aspects of body image (cognitive, affective, behavioral or perceptual aspects). Above all, most studies assessed patients through the first 2 years post bariatric surgery while weight regain is observed after the initial post surgical period [6]. The aim of our study was to describe the evolution of body image in a prospective observational cohort for up to 5 years after BS.

## Materials and methods

We carried out a single-center prospective observational cohort study in the Obesity Department of Bordeaux University Hospital between October 2011 to April 2012.The inclusion criteria required patients to be over 18 years of age, to speak and to write French and undergoing a preoperative assessment for BS according to international guidelines [16]. Patients undergoing bariatric surgery during this period and who agreed to participate in the study were included. Data were collected during the preoperative assessment, and between 1 and 3 months (M1–M3), 6 and 9 months (M6–M9), 12 and 18 months (M12–M18), and 60 and 72 months (M60–M72) after BS. Body weight, height, body mass index (BMI), and percentage of weight lost after BS were assessed at each of these time points.

The Multidimensional Body-Self Relations Questionnaire-Appearance Scale (MBSQR-AS) [17] and the Stunkard Figure Rating Scale were used to assess body image satisfaction.

The MBSQR-AS assesses self-attitudinal aspects of the body-image construct and is composed of 34 questions divided into five sections: 1) 'appearance evaluation' measures feelings of physical attractiveness or unattractiveness, with a high score indicating satisfaction with one's own appearance; 2) 'appearance orientation' measures the extent of investment in one's appearance. A high score indicates a greater investment; 3) 'body areas satisfaction scale' (BASS), where a high score indicates general satisfaction with most areas of one's body and a low score indicates dissatisfaction in the size or appearance of several areas; 4) 'overweight

preoccupation' measures fat anxiety, weight vigilance, dieting, and eating restraint; and 5) 'self-classified weight' measures how one perceives his/her weight, from very underweight to very overweight. The MBSRQ–AS has been translated and validated in French [18]. The five subscales of the MBSRQ–AS have good psychometric qualities: the internal consistency ranges from .66 to .88 and test-retest reliability ranges from .78 to .85 for the five subscales in French. The scale has been used in numerous studies focusing on obesity and bariatric surgery.

The Stunkard Figure Rating Scale [19] presents a series of nine male and female schematic silhouettes ranging from skinny to obese. Participants are asked to choose the silhouette that corresponds to their ideal body size (IBS) and the silhouette that reflects their current body size (CBS). Body dissatisfaction (BD) is calculated by subtracting the ideal body size from the current body size. A positive score indicates a desire to be thinner, while a negative score indicates a desire to be heavier.

Statistical analyses were performed using GraphPadPrism software. Quantitative variables were compared by an analysis of variance (ANOVA). Correlation coefficients (Spearman or Pearson coefficient, depending on the distribution of continuous or non-continuous variables) were calculated. A $P$-value$< 0.05$ was considered statistically significant. All quantitative variables are expressed as the mean ± standard error of the mean (SEM).

Each participant received a study information sheet and signed an informed consent form.

## Results

### Patient baseline characteristics

Sixty one white patients (female, n = 47, (77%)) were included in the study. Their mean age was 45 ± 10 years. The average baseline bodyweight was 125.6 ± 22.7 kg corresponding to an average BMI of 42.4 ± 7.9 kg/m$^2$. A Roux-en-Y Gastric Bypass (RYGB) was performed in 32 patients (52%) while 29 patients (48%) underwent a sleeve-gastrectomy.

At baseline the mean scores for the MBSQR-AS sections were as follows: appearance evaluation = 2.30 ± 0.66, appearance orientation = 3.58 ± 0.56, BASS = 2.44 ± 0.51, overweight preoccupation = 3.29 ± 0.68, and self-classified weight = 4.58 ±0.60. The average anxiety score was 7.43 ± 3.42, and the mean depression score was 4.97 ± 3.17. The average BD score from the Stunkard Figure Rating Scale was 3.30 ± 1.01. The patients' characteristics are summarized in Table 1.

**Table 1. Characteristics of the bariatric surgery candidates at the preoperative assessment.** Quantitative variables are expressed as the mean ± SEM.

| | |
|---|---|
| **Men/women (%)** | 23/77 |
| **Mean age (years)** | 45 ± 10 |
| **Mean weight (kg)** | 125.6 ± 22.7 |
| **Mean BMI (kg/m$^2$)** | 42.4 ± 7.9 |
| **Onset of obesity** | |
| Since childhood (%) | 54 |
| Since adulthood (%) | 46 |
| **Surgery type:** | |
| Bypass (%) | 52 |
| Sleeve-gastrectomy (%) | 48 |

BMI = Body Mass Index.

## Body weight loss after BS

No patients were lost to follow-up until 18 months after BS. At M60–M72, 21 patients (34%) were lost to follow-up (12 dropouts and 9 missing answers on the questionnaires).

The percentage of body weight loss increased significantly fromM1–M3 (10.62 ± 4.40) until M12–M18 (28.44 ± 7.99%; $P < 0.05$) before decreasing significantly to 22.82 ± 12.05% at M60–M72 ($P < 0.05$). The average BMI decreased significantly from the preoperative assessment (39.59 ± 7.34 kg/m$^2$)until M12–M18 (M1–M3 = 33.80 ± 6.69 kg/m$^2$, M6–M9 = 33.80 ± 6.69 kg/m$^2$and M12–M18 = 31.63 ± 6.38 kg/m$^2$; $P < 0.05$), but increased significantly between M12–M18(31.63 ± 6.38 kg/m$^2$) and M60–M72 (34.12 ± 8.22 kg/m$^2$;$P < 0.05$).

## Changes in MBSRQ-AS scores during follow-up

The appearance evaluation scores increased significantly from the preoperative period (2.30 ± 0.66) to the M12–M18 assessment (3.15 ± 0.78;$P < 0.05$) and remained significantly elevated through M60–M72 (Table 2). There were positive correlations between weight loss and appearance evaluation at M1–M3 (r = 0.44; $P < 0.05$), M6–M9 (r = 0.42; $P < 0.05$), and M60–M72 (r = 0.33; $P < 0.05$).

The appearance orientation scores increased significantly fromM1–M3 to M6–M9 but decreased significantly between M12–M18 and M60–M72. At that last time point scores were not statistically different from the pre-surgery scores. There was no correlation between weight loss and appearance orientation during follow-up.

The BASS scores increased significantly from the preoperative assessment to M6-M9 (2.44 ± 0.5 vs 2.76 ± 0.56 vs 3.05 ± 0.61, preoperative assessment vs M1-M3 vs M6-M9, $P < 0.05$). The scores did not change between M6–M9 and M12–M18 (3.24 ± 0.64) but decreased significantly until M60–M72 (2.90 ± 0.74), though they remained significantly higher than at the preoperative evaluation (Table 2). There were positive correlations between weight loss and BASS at M1–M3(r = 0.4, $P < 0.05$), M6–M9 (r = 0.41, $P < 0.05$), M12–M18 (r = 0.39, $P < 0.05$), and M60–M72 (r = 0.36, $P < 0.05$).

The overweight preoccupation scores did not change significantly until M6–M9 (preoperative = 3.29 ± 0.68, M1–M3 = 3.18 ± 0.67, and M6–M9 = 3.19 ± 0.75;$P < 0.05$). Between M6-M9 and M12–M18, the score decreased significantly and did not change until M60–M72 (Table 2). There was a negative correlation between weight loss and overweight preoccupation at M1–M3 (r = -0.39, $P < 0.05$). The self-classified weight scores decreased significantly from M1–M3 (4.43 ± 0.71) to M6–M9 (3.90 ± 0.76) and M12–M18 (3.67 ± 0.72;$P < 0.05$). These scores increased at M60–M72 (3.96 ± 0.63) but remained lower than the preoperative scores (4.58 ± 0.60). There was no correlation between weight loss and self-classified weight score during follow-up. Changes in the MBSRQ-AS scores are summarized in Table 2.

At last follow-up, the appearance evaluation was significantly higher in patients that had achieved a BMI <30 kg /m2 compared to those who did not (3.44 ± 0.65 vs 2.83 ± 0.79). There

**Table 2. The MBSRQ-AS scores (mean ± SEM) during follow-up.** Means with different superscripts are significantly different from each other ($P < 0.05$).

| | Preoperative assessment | M1–M3 | M6–M9 | M12–M18 | M60–M72 |
|---|---|---|---|---|---|
| **Appearance evaluation** | 2.30 ± 0.66 | 2.59 ± 0.72[a] | 3.04 ± 0.76[b] | 3.15 ± 0.78[c] | 2.97 ± 0.80[abc] |
| **Appearance orientation** | 3.58 ± 0.56[b] | 3.63 ± 0.55[b] | 3.81 ± 0.53[a] | 3.81 ± 0.50[ab] | 3.63 ± 0.53[b] |
| **BASS** | 2.44 ± 0.51 | 2.76 ± 0.56[b] | 3.05 ± 0.61[a] | 3.24 ± 0.64[ac] | 2.90 ± 0.74[bc] |
| **Overweight preoccupation** | 3.29 ± 0.68[ab] | 3.18 ± 0.67[ab] | 3.19 ± 0.75[a] | 3.06 ± 0.62[b] | 2.96 ± 0.70[ab] |
| **Self-classified weight** | 4.58 ± 0.60[a] | 4.43 ± 0.71[a] | 3.90 ± 0.76[b] | 3.67 ± 0.72 | 3.96 ± 0.63[b] |

BASS = Body Areas Satisfaction Scale.

**Table 3. The BD score (mean ± SEM) during follow-up.** Means with different superscripts are significantly different from each other ($P < 0.05$).

|  | Preoperative assessment | M1–M3 | M6–M9 | M12–M18 | M60–M72 |
|---|---|---|---|---|---|
| **BD score** | 3.30 ± 1.01 | 2.61 ± 1.04 | 1.85 ± 0.95[a] | 1.33 ± 0.96[b] | 1.77 ± 1.37[ab] |

BD = Body Dissatisfaction.

was no differences between patients with BMI under or over 30 for the other scores that composed the MBSRQ-AS score.

## Changes in BD score during follow-up

The mean BD score decreased significantly between the preoperative assessment (3.30 ± 1.01), M1–M3 (2.61 ± 1.04), M6–M9 (1.85 ± 0.95), and M12–18 (1.33 ± 0.96; $P < 0.05$). The mean BD score at M60–M72 was 1.77 ± 1.37 (Table 3). There was a significant negative correlation between weight loss and BD score at M60–M72 only (r = -0.47, $P < 0.05$). Changes in the BD scores are summarized in Table 3.

## Discussion

While bariatric surgery is by far the most effective treatment for long-term obesity and its comorbidities, less is known about body image following BS especially after the first 2 years after BS. The current analysis, which includes a longitudinal characterization of body image through 5 years provides interesting results about the durability of body image changes after BS. In our study, the appearance evaluation and self-classified weight scores improved until M12–M18 after BS and declined at M60-72 but remained higher than the preoperative scores. These results have already been highlighted in the literature although the follow-up is most often of shorter duration and most studies have a post-operative sample size of fewer than 50 individuals. Indeed, in the systematic review of the literature of Ivezaj et al. [12], out of the 31 observational longitudinal studies, 52% (n = 16) had up to 12 months of follow-up and only 6% (n = 2) up to 48 months. All the studies have shown an improvement in body image after bariatric surgery and some studies have already shown stabilization or a decline of satisfaction scores from 12 or 24 months [20–22]. The studies with the longest follow-up agree that the satisfaction scores stay higher 48 months after BS than before surgery [23–25].

These results have to be faced to the evolution of the BASS. Therefore, the BASS scores only improved until M6–M9 reflecting dissatisfaction with one or more body regions. This could be linked to the appearance of excessive skin that is quite common after massive weight [26, 27]. Body contouring surgery (BCS) could improve body image in these patients [28]. However little is known about the trajectory of body image post-BCS among bariatric surgery patients. BCS after BS seems to improve body image satisfaction only in areas involved in the surgery (not untreated areas) [29] while one longitudinal study suggests that body image satisfaction needs three or more plastic surgeries following BS [30] to score comparable to standards. Only 8 patients from our cohort had undergone BCS after BS but the different types of surgery, the different delays between BS and BCS and short-term follow-up did not allow us to analyze their data.

This dynamic change of body image according to time after BS mirrors the weight loss trajectories that happen after BS [31]. Therefore, viewing post BS events (not only weight outcome) as a dynamic process is more a realistic view of post BS outcomes. This also emphasizes the need for a careful and long-life follow-up after BS. This follow-up needs to include body weight, nutritional status but also body image. Indeed, in non-surgical population, various

features of body image dissatisfaction may be a signal for greater eating-disorder psychopathology, which could ultimately impact long-term weight loss outcomes following bariatric surgery [32, 33].

The literature on post-operative body image changes and weight outcome is mixed. Consistent with findings from some previous studies [34, 35], we found a positive correlation between weight loss and appearance evaluation at M1-M3, M6-M9 and M60-M72. There were also positive correlations between weight loss and BASS at all time points between M1-M3 and M60-M72. These positive associations are likely to be of clinical importance. Therefore, improving these dimensions could increase the amount of weight lost after BS. Targeted psychoeducation and assessment of body image throughout the bariatric surgery process both before and after bariatric surgery is mandatory to optimize body weight loss after BS [36].

The overweight preoccupation and appearance orientation scores were only slightly modified during follow-up and the average score at 5 years was not significantly different from the preoperative average. The lack of improvement in this aspect of body image has already been described in the literature [22, 23, 37]. This indicates that body weight remains a major concern for BS patients even after significant weight loss [38, 39]. Indeed, these scores remained positive throughout the follow-up period, suggesting: i) a persistent desire of patients to be thinner than they are, ii) that their perceived body remains different from their ideal body, and iii) that they constantly see themselves as heavy even after significant weight loss. These results are similar to those of previous studies [40, 41], although one study showed a significant reduction in the determination for weight loss at 6 months post-BS [42].

These results suggest that patients may still identify as obese, even after BS, as indicated by their tendency to overestimate their weight [43]. In patients suffering from anorexia nervosa, weight overestimation is well described and may correlate with dysfunction of the parietal cortex [44]. A similar mechanism has been suggested after rapid weight loss in BS patients where proprioceptive information is not updated, which leads to a distorted body schema. Consequently, patients feel heavier than they actually are [45].

Although we can't claim causality between body weight loss and body image, we suppose that the decline of body image satisfaction long after bariatric surgery should prompt clinicians to think about ways to improve body image. Virtual reality could be an option. Indeed, according to the 'allocentric lock hypothesis', patients suffering from obesity may approach their body image from an allocentric frame of reference by adopting the point of view of an outside observer toward their own body, and this point of view is not updated after weight loss [46–48]. Memories of stigmatizing experiences can contribute to allocentric locking. The inability to update the allocentric representation therefore locks the patient into a dissatisfying body. Despite efforts to modify their weight, they will always be present in a body that differs from reality [49, 50]. Two case series showed significant improvements in the satisfaction scores of BS patients after 6 weeks of virtual reality [51, 52]. However, more studies and longer follow-up periods are needed to clarify the impact of this practice.

To our knowledge, it is the only study with so many body image assessments 5 years after bariatric surgery [53, 54]. The duration of our study allowed us to detect changes between 1 and 5 years after BS. Interestingly, we studied individuals from the same ethnic background that can potentially influence body image [55]. To our knowledge, this element has never been studied after bariatric surgery. In addition, we used 2 measures of body image.

However, none of the questionnaires we used were specific to BS. However, this is also the case for most published studies in this field. Indeed, a recent review identified only six questionnaires specific to assessing body image in patients after BS [56]. The BODY-Q questionnaire, developed specifically in 2012 for BS patients, seems to be the most appropriate for addressing these specific issues [57]. However, none of these questionnaires are translated or

validated in French. The Figure Rating Scale has many limitations: limited range of figures are presented on the scale (it is indeed not a tool for subjects with super obesity), illustrations may not capture all the different body types, and validity may vary by ethnic group. Nevertheless, this tool was used frequently when we started collecting data and is still used today [58]. In addition, it has the advantage of being quickly administered, showing a strong correlation with the body mass index. It provides a simple, self-administered technique of collecting estimates of body dissatisfaction, and avoids variability that might be caused by language deficits. An interesting option could have been the use of the Body-Image Assessment for obesity or BIA-O as a figural rating scale (the first one adapted specifically for use in obesity) [59].

Twenty-one patients (34%) were lost to follow-up during the study. This difficult observance of follow-up after bariatric surgery is indeed well described in the literature and multiple factors have been suggested as risk factors for this attrition (distance to travel to the clinic, younger age, unemployment, financial factors, psychological issues) [60]. Of note, our data did not allow us to study the relationships between post-BS body image changes and factors associated with greater body image dissatisfaction in obese patients such as early-onset obesity, episodes of obesity-related stigma, eating disorders, underlying depression, yo-yo dieting, and overestimating weight.

## Conclusions

Our data show that body image improves after BS, but this effect is only temporary. The use of questionnaires specific to BS, including the BODY-Q, would help us better understand this change over time.

## Supporting information

**S1 File.**
(XLS)

## Acknowledgments

We thank the Bordeaux University Hospital which allowed this work. We thank the subjects who participated to this work. The English in this document has been checked by at least two professional editors, both native speakers of English. For a certificate, please see: http://www.textcheck.com/certificate/AbjS4I

## Author Contributions

**Conceptualization:** Laurène Bosc, Blandine Gatta-Cherifi.

**Data curation:** Laurène Bosc.

**Supervision:** Blandine Gatta-Cherifi.

**Validation:** Flore Mathias, Maud Monsaingeon, Caroline Gronnier, Emilie Pupier.

**Writing – original draft:** Laurène Bosc, Blandine Gatta-Cherifi.

**Writing – review & editing:** Laurène Bosc, Blandine Gatta-Cherifi.

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
