## [Decision Letter · Decision Letter 0]

15 Mar 2022

PONE-D-21-30163

Long-term changes in body image after bariatric surgery: an observational cohort study

PLOS ONE

Dear Dr. Gatta-Cherifi,

Thank you for submitting your manuscript to PLOS ONE. After careful consideration, we feel that it has merit but does not fully meet PLOS ONE’s publication criteria as it currently stands. Therefore, we invite you to submit a revised version of the manuscript that addresses the points raised during the review process.

We look forward to receiving your revised manuscript.

Kind regards,

Anca Mihaela Pantea-Stoian, md, phd

Academic Editor

PLOS ONE

3. In your Methods section, please provide a justification for the sample size used in your study, including any relevant power calculations (if applicable).

Furthermore, please amend your manuscript to adhere to our submission guidelines with respect to language describing demographic groups. Outmoded terms and potentially stigmatizing labels should be changed to more current, acceptable terminology. Specifically, we recommend that “Caucasian” should be changed to more appropriate term(s) such as 'White'

4. PLOS requires an ORCID iD for the corresponding author in Editorial Manager on papers submitted after December 6th, 2016. Please ensure that you have an ORCID iD and that it is validated in Editorial Manager. To do this, go to ‘Update my Information’ (in the upper left-hand corner of the main menu), and click on the Fetch/Validate link next to the ORCID field. This will take you to the ORCID site and allow you to create a new iD or authenticate a pre-existing iD in Editorial Manager. Please see the following video for instructions on linking an ORCID iD to your Editorial Manager account: https://www.youtube.com/watch?v=_xcclfuvtxQ.

Additional Editor Comments:

Dear Authors,

The article is interesting and valuable. However, there are some improvements to make:

1. Line 56: The authors mention that bariatric surgery also significantly improves most obese patients' comorbidities. Could you please detail (for example, diabetes, metabolic syndrome, hypertension..)?

2. It would be interesting if you could make some correlations between several factors and the change in body image over time, such as sex, age, the onset of obesity, type of BS, comorbidities, weight loss.

3. Do you have separated the patients who achieved BMI <30? It could show different results between the groups. It is interesting to explain the results, for this cluster of patients .

4. Please explain and extend the discussion about the reasons for the dropouts after 18 months.

5. Refresh the whole manuscript, correct miswriting, and check the punctuation.

Reviewers' comments:

Reviewer's Responses to Questions

**Comments to the Author**

1. Is the manuscript technically sound, and do the data support the conclusions?

Reviewer #1: Yes

Reviewer #2: Yes

2. Has the statistical analysis been performed appropriately and rigorously? 

Reviewer #1: Yes

Reviewer #2: Yes

3. Have the authors made all data underlying the findings in their manuscript fully available?

Reviewer #1: Yes

Reviewer #2: Yes

4. Is the manuscript presented in an intelligible fashion and written in standard English?

Reviewer #1: No

Reviewer #2: Yes

5. Review Comments to the Author

Reviewer #1: Dear Authors,

The aim of following patients' body images for a longer period after bariatric surgery is a very important point considering it may help to build strategies to achieve better and prolonged results, even if not using the most appropriate tools. Just wondering if you had separated the patients who achieved BMI <30 it could show different results between the groups. Also, I wondered if wouldn't be interesting to refer to the reasons for the dropouts after 18 months.

I suggest you review the writing as there are a few miswriting and check the punctuation.

Reviewer #2: The paper is an original article regarding the body image perception and its variation in patients that underwent bariatric surgery. The topic is extremely interesting. The paper is written in a fluid English language and based on a solid statistical analysis. Some issues could be improved before publication:

1. Line 56: The authors mention that bariatric surgery also bring important improvement in comorbidities that most obese patients have. Could you please detail (for example: diabetes, metabolic syndrom, hypertension..)?

2. It would be interesting if you could make some correlations between several factors and the change in body image over time, such as sex, age, onset of obesity, type of BS, comorbidities, weight loss

6. PLOS authors have the option to publish the peer review history of their article (what does this mean?). If published, this will include your full peer review and any attached files.

Reviewer #1: No

Reviewer #2: No

---

## [Author Response · Author response to Decision Letter 0]

11 May 2022

Please see below for a point-by-point response to the comments and concerns. All page numbers refer to the revised manuscript file with tracked changes.

Additional Editor Comments:

Dear Authors,

The article is interesting and valuable. However, there are some improvements to make:

1. Line 56: The authors mention that bariatric surgery also significantly improves most obese patients' comorbidities. Could you please detail (for example, diabetes, metabolic syndrome, hypertension..)? 

We thank the reviewer for this pertinent comment. We detailed that in the manuscript.

2. It would be interesting if you could make some correlations between several factors and the change in body image over time, such as sex, age, the onset of obesity, type of BS, comorbidities, weight loss.

We thank the reviewer for this comment. We did not find any correlation between the factors you mentioned above and the change in body image over time.

3. Do you have separated the patients who achieved BMI <30? It could show different results between the groups. It is interesting to explain the results, for this cluster of patients .

As you requested, we had a look at patients according to their final BMI i.e > 30 or < 30 and we added a sentence in the manuscript (lines 171-174).

4. Please explain and extend the discussion about the reasons for the dropouts after 18 months.

As you suggested, we explained that in the discussion (lines 273-276).

5. Refresh the whole manuscript, correct miswriting, and check the punctuation.

We checked the whole manuscript to ensure that the text is optimally phrased and free from typographical and grammatical errors.

Reviewer #1: Dear Authors,

The aim of following patients' body images for a longer period after bariatric surgery is a very important point considering it may help to build strategies to achieve better and prolonged results, even if not using the most appropriate tools. Just wondering if you had separated the patients who achieved BMI <30 it could show different results between the groups. Also, I wondered if wouldn't be interesting to refer to the reasons for the dropouts after 18 months.

I suggest you review the writing as there are a few miswriting and check the punctuation.

Reviewer #2: The paper is an original article regarding the body image perception and its variation in patients that underwent bariatric surgery. The topic is extremely interesting. The paper is written in a fluid English language and based on a solid statistical analysis. Some issues could be improved before publication:

1. Line 56: The authors mention that bariatric surgery also bring important improvement in comorbidities that most obese patients have. Could you please detail (for example: diabetes, metabolic syndrom, hypertension..)?

We thank the reviewer for this pertinent comment. We detailed that in the manuscript.

2. It would be interesting if you could make some correlations between several factors and the change in body image over time, such as sex, age, onset of obesity, type of BS, comorbidities, weight loss

As explained before, we did not find any correlation between the factors you mentioned above and the change in body image over time.

---

## [Decision Letter · Decision Letter 1]

11 Aug 2022

PONE-D-21-30163R1Long-term changes in body image after bariatric surgery: an observational cohort studyPLOS ONE

Dear Dr. Gatta-Cherifi,

Thank you for submitting your manuscript to PLOS ONE. After careful consideration, we feel that it has merit but does not fully meet PLOS ONE’s publication criteria as it currently stands. Therefore, we invite you to submit a revised version of the manuscript that addresses the points raised during the review process.

Your manuscript has been seen by two additional reviewers and their comments are attached below. We would like to ask you to address the concerns of reviewer #3 and respond to the comments of reviewer #4, specifically:please discuss the limitation regarding the Figure Rating Scaleplease review current literature to ensure that the most recent literature on the topic has been includednote that citing the reference requested by Reviewer 4 is not a requirement for publicationCould you please revise your manuscript to include their concerns?

We look forward to receiving your revised manuscript.

Kind regards,

Thomas Tischer

Staff Editor

PLOS ONE

Journal Requirements:

Reviewers' comments:

Reviewer's Responses to Questions

**Comments to the Author**

1. If the authors have adequately addressed your comments raised in a previous round of review and you feel that this manuscript is now acceptable for publication, you may indicate that here to bypass the “Comments to the Author” section, enter your conflict of interest statement in the “Confidential to Editor” section, and submit your "Accept" recommendation.

Reviewer #1: All comments have been addressed

Reviewer #2: All comments have been addressed

Reviewer #3: (No Response)

Reviewer #4: All comments have been addressed

2. Is the manuscript technically sound, and do the data support the conclusions?

Reviewer #1: Yes

Reviewer #2: Yes

Reviewer #3: No

Reviewer #4: Yes

3. Has the statistical analysis been performed appropriately and rigorously? 

Reviewer #1: Yes

Reviewer #2: I Don't Know

Reviewer #3: No

Reviewer #4: Yes

4. Have the authors made all data underlying the findings in their manuscript fully available?

Reviewer #1: Yes

Reviewer #2: Yes

Reviewer #3: Yes

Reviewer #4: Yes

5. Is the manuscript presented in an intelligible fashion and written in standard English?

Reviewer #1: Yes

Reviewer #2: Yes

Reviewer #3: Yes

Reviewer #4: Yes

6. Review Comments to the Author

Reviewer #1: (No Response)

Reviewer #2: The authors responded to the queries in the revised version. I have no further issues. I recommend publication.

Reviewer #3: This manuscript reports on data that is almost a decade old, for reasons that are unclear. Since that time, research in this area has grown to a size larger than suggested here. Much of the referenced literature is dated and many more recent empirical papers and reviews on this topic were missed. Further, the Figure Rating Scale is no longer viewed as an appropriate measure of body image dissatisfaction. It certainly not validated for those with clinically severe obesity. The results of the MBSRQ-AS replicate those from other studies. So, it's not clear what this small, dated study adds to the literature.

Reviewer #4: Long-term changes in body image after bariatric surgery: an observational cohort study.

Bariatric surgery and reconstructive procedures after weight loss are one of the most important and popular surgeries nowadays. On the other hand, the problem lies not only in the body structure and obesity. There is very an important, maybe the most important part how to prepare our patients’ mentally for the surgical journey. The surgery itself may give us a false belief that the patient is treated. But the psychiatric part may be as important as 80% of the treatment. It would be interesting to focus on the surgical outcome in comparison with the mental preparation before the surgery.

I find the study interesting. The manuscript is well organized, and statistically well prepared. Maybe the idea is not new, and the outcome is similar to the other studies, even the one presented by my team in 2020 The long-term effect of body contouring procedures on the quality of life in morbidly obese patients after bariatric surgery

doi.org/10.1371/journal.pone.0229138.

The strength of the study is long followed for up to 5 years. The response rate of the study is 66% which is more than needed to assess as a valid study. One of the most important parts of the manuscript is to show that after 5 years there is a decline in the body assessment, but it is still higher than before bariatric surgery. So we can say that the patient qualification process was proper.

There is my summary

1. The authors have responded properly to the reviewers

2. The study is valuable due to the long follow up

3. The response rate is 66% - enough to publish

4. Would be better to have a multicenter study, but thanks to long follow-up it should be considered valuable.

5. Please add the mentioned study

6. In the next studies I would recommend the assessment of the body reception and psychiatric evaluation of the patients.

I recommend this manuscript be accepted as an original article, please add the publication I mentioned.

7. PLOS authors have the option to publish the peer review history of their article (what does this mean?). If published, this will include your full peer review and any attached files.

Reviewer #1: No

Reviewer #2: No

Reviewer #3: No

Reviewer #4: **Yes: **Marek A Paul

---

## [Author Response · Author response to Decision Letter 1]

20 Sep 2022

Dear Editor and Reviewers, 

Thank you for giving us the opportunity to submit another revised version of the manuscript “Long-term changes in body image after bariatric surgery: an observational cohort study” for publication in PLOS ONE. 

We appreciate the time and effort that you and the reviewers dedicated to providing feedback on our manuscript and are grateful for the insightful comments on and valuable improvements to our paper. We have incorporated most of the suggestions made by the reviewers. Those changes are highlighted within the manuscript. 

Please see below for a point-by-point response to the comments and concerns. All page numbers refer to the revised manuscript file with tracked changes.

Reviewer #3: This manuscript reports on data that is almost a decade old, for reasons that are unclear. Since that time, research in this area has grown to a size larger than suggested here. Much of the referenced literature is dated and many more recent empirical papers and reviews on this topic were missed. Further, the Figure Rating Scale is no longer viewed as an appropriate measure of body image dissatisfaction. It certainly not validated for those with clinically severe obesity. The results of the MBSRQ-AS replicate those from other studies. So, it's not clear what this small, dated study adds to the literature.

Response:

Thank you for taking the time to review our article.

Indeed, our data are almost a decade ago since one of our goal was to follow patients during 5 years that means that the study started almost a decade ago.

We agree that the Figure Rating Scale has many limitations: limited range of figures are presented on the scale (it is indeed not a tool for subjects with super obesity), illustrations may not capture all the different body types, and validity may vary by ethnic group. Nevertheless, this tool was used frequently when we started collecting data and is still used today [58].

In addition, it has the advantage of being quickly administered, showing a strong correlation with the body mass index. It provides a simple, self-administered technique of collecting estimates of body dissatisfaction, and avoids variability that might be caused by language deficits. 

An interesting option could have been the use of the Body-Image Assessment for obesity or BIA-O as a figural rating scale (the ﬁrst one adapted speciﬁcally for use in obesity) with the BODY-Q as already mentioned in the manuscript [59]. The BIA-O was however much less used than the FRS in the bariatric surgery literature.

We think that the current research could add to the literature because it is the only study with so many body image assessments 5 years after bariatric surgery. We added this information in the manuscript [53, 54].

We thank the reviewer for giving us the opportunity to read again the literature. We have now cited additional recent studies in our paper.

[28] Paul MA, Opyrchał J, Knakiewicz M, Jaremków P, Duda-Barcik Ł, Ibrahim AMS, Lin SJ. The long-term effect of body contouring procedures on the quality of life in morbidly obese patients after bariatric surgery. PLoS One. 2020 Feb 21;15(2):e0229138.

[37] Perdue TO, Schreier A, Swanson M, Neil J, Carels R. Majority of female bariatric patients retain an obese identity 18-30 months after surgery. Eat Weight Disord. 2020 Apr;25(2):357-364. 

[53] Mento C, Silvestri MC, Muscatello MRA, Rizzo A, Celebre L, Cedro C, Zoccali RA, Navarra G, Bruno A. The role of body image in obese identity changes post bariatric surgery. Eat Weight Disord. 2022 May;27(4):1269-1278

[54] Bertoletti J, Galvis Aparicio MJ, Bordignon S, and Marceli Trentini C. Body Image and Bariatric Surgery: A Systematic Review of Literature.Bariatric Surgical Practice and Patient Care. Jun 2019.81-92.

[58] Parzer V, Sjöholm K, Brix JM, Svensson PA, Ludvik B, Taube M. Development of a BMI-Assigned Stunkard Scale for the Evaluation of Body Image Perception Based on Data of the SOS Reference Study. Obes Facts. 2021; 14(4):397-404.

[59] Williamson DA, Womble LG, Zucker NL, Reas DL, White MA, Blouin DC, Greenway F. Body image assessment for obesity (BIA-O): development of a new procedure. Int J Obes Relat Metab Disord. 2000 Oct;24(10):1326-32.

Reviewer #4: 

I find the study interesting. The manuscript is well organized, and statistically well prepared. Maybe the idea is not new, and the outcome is similar to the other studies, even the one presented by my team in 2020 the long-term effect of body contouring procedures on the quality of life in morbidly obese patients after bariatric surgery

doi.org/10.1371/journal.pone.0229138.

The strength of the study is long followed for up to 5 years. The response rate of the study is 66% which is more than needed to assess as a valid study. One of the most important parts of the manuscript is to show that after 5 years there is a decline in the body assessment, but it is still higher than before bariatric surgery. So we can say that the patient qualification process was proper.

There is my summary

1. The authors have responded properly to the reviewers

2. The study is valuable due to the long follow up

3. The response rate is 66% - enough to publish

4. Would be better to have a multicenter study, but thanks to long follow-up it should be considered valuable.

5. Please add the mentioned study

6. In the next studies I would recommend the assessment of the body reception and psychiatric evaluation of the patients.

Response :

Thanks for your comment. Your study using the BODY-Q is very interesting and we have added your reference to our manuscript at line 211 [28].

---

## [Editor Report · Decision Letter 2]

2 Oct 2022

Long-term changes in body image after bariatric surgery: an observational cohort study

PONE-D-21-30163R2

Dear Dr. Gatta-Cherifi,

We’re pleased to inform you that your manuscript has been judged scientifically suitable for publication and will be formally accepted for publication once it meets all outstanding technical requirements.

Kind regards,

George Vousden

Staff Editor

PLOS ONE
---

## [Editor Report · Acceptance letter]

9 Nov 2022

PONE-D-21-30163R2 

Long-term changes in body image after bariatric surgery : an observational cohort study 

Dear Dr. Gatta-Cherifi:

I'm pleased to inform you that your manuscript has been deemed suitable for publication in PLOS ONE. Congratulations! Your manuscript is now with our production department. 

Kind regards, 

on behalf of

Dr. George Vousden 

Staff Editor

PLOS ONE